# Factors influencing preoperative chest radiography request for elective endoscopic procedures among medical personnel

Pawit Somnuke[1], Rachaneekorn Ramlee[2], Waratchaya Ratanapaiboon[3], Passorn Thommaaksorn[1], Cherdsak Iramaneerat[4], Somsit Duangekanong[5], Arunotai Siriussawakul[1,2]*

1 Department of Anesthesiology, Faculty of Medicine, Siriraj Hospital, Mahidol University, Bangkok, Thailand, 2 Integrated Perioperative Geriatric Excellent Research Center, Faculty of Medicine, Siriraj Hospital, Mahidol University, Bangkok, Thailand, 3 Department of Anesthesiology, Veterans General Hospital, Bangkok, Thailand, 4 Department of Surgery, Faculty of Medicine, Siriraj Hospital, Mahidol University, Bangkok, Thailand, 5 Graduate School of Advanced Technology Management, Faculty of Information Technology and Management, Assumption University, Bangkok, Samuthprakarn, Thailand

* arunotai.sir@mahidol.ac.th

**Data Availability Statement:** All relevant data are within the paper and its Supporting information files.

## Abstract

### Background

Chest radiography is not routinely recommended before elective endoscopies. A high incidence of perioperative chest radiography requests was observed at our institution. This study aims to investigate factors influencing preoperative chest radiography request for patients undergoing elective gastrointestinal (GI) endoscopies.

### Methods

This cross-sectional clinical study recruited 264 participants from different medical specialties who were responsible for preoperative endoscopic chest x-ray (CXR) ordering including anesthesiologists, surgeons and gastroenterologists. They completed questionnaires exploring their general knowledge and attitudes about preoperative chest radiography. Demographic characteristic of the participants affecting the knowledge on preoperative chest radiography was determined. A Structural Equation Model (SEM) was constructed from validated conceptual framework to find causal relationships between hypothesized factors and intention for preoperative endoscopic chest radiography request. Statistical analyses were performed using the SPSS software version 18.0 and Analysis of Moment Structures (AMOS) version 18.0.

### Results

The questionnaire response rate was 53.79%. Baseline general knowledge on preoperative chest radiography of the participants was comparable. The SEM results showed unsupported relationship between hypothesized factors and the intention for preprocedural GI endoscopic CXR request (p < 0.1).

**Funding:** This research project was supported by the Faculty of Medicine, Siriraj Hospital, Mahidol University, grant number [IO] R016131009 on 29 November 2017 to AS. The funders had no role in study design, data collection and analysis, decision to publish, or preparation of the manuscript.

**Competing interests:** The authors have declared that no competing interests exist.

## Conclusions

General knowledge of medical personnel on tuberculosis needs improvement. To rectify the unnecessary chest radiography request before elective GI endoscopic procedures, awareness of the patients' health conditions, adherence to the hospital's policy and realizing of possible patient-related mishaps are not the determinants for preprocedural endoscopic chest radiography request. Future works are required to explore other alternative factors involved for reducing chest radiography requests which are not indicated.

## Introduction

Preoperative assessments are considered prerequisite for patients undergoing surgical procedures in elective or emergency settings. Adequate tests are helpful in identifying and determining risks, optimizing anesthetic techniques to reduce morbidity and mortality, directing postoperative management, and preventing prosecutions in case there are any adverse events during the operations [1]. However, preoperative screening can be costly, resource intensive, time-consuming, and uncomfortable for patients. Ideally, the ordering of preprocedural testing should be based on data gathered from patients, comprising their histories, comorbidities, and significant findings from physical examination (American Society for Gastrointestinal Endoscopy (ASGE) guidelines [2]. Notwithstanding the paramount need for preoperative testing, several studies have concluded that no benefit is provided by routine testing in cases of elective, low-risk, ambulatory surgery for either adult or pediatric patients [3, 4].

Chest radiography is one of the frequent paradigms of preoperative tests covered by recent preoperative guidelines for elective surgery. It is recommended for patients over 60 years of age, especially those with a history of smoking, an American Society of Anesthesiologists (ASA) physical status of 3 or higher, a respiratory tract infection, signs and symptoms of cardiopulmonary disease, and decompensated heart failure [5–8]. In contrast, chest radiography is not routinely recommended before certain elective surgeries, especially endoscopies, because the incidence of radiographic figures that alters clinical outcomes is as low as 0.1% among all abnormal images [9, 10] and the endoscopic procedures themselves are regarded as low risk, having a cardiac event risk of less than 1%. Previous studies reported that there were insufficient data to determine the benefits of routine laboratory testing before endoscopic procedures; nevertheless, surgeons were inclined to unnecessarily request routine laboratory and preoperative screening tests [6, 11–14]. Less than one per cent of tests from all patients have been reported to reveal abnormalities that could affect perioperative outcomes [15]. Routine testing in cases of low-risk surgery may result in extra testing, exposure to radiation, surgery cancellation, increased patient anxiety, and additional hospital expenses [4, 5, 16, 17]. In our institution, Siriraj Hospital, approximately 78% of the patients scheduled for non-neurological and non-cardiovascular-thoracic operations were proceeded to preoperative chest radiography [18]. In addition, from the chart reviews of the patients scheduled for elective gastrointestinal endoscopic procedures in Siriraj Hospital, 52.1% were reported with abnormal chest radiography which cardiomegaly predominated. However, active pulmonary lesion accounted for only 0.2% [19, 20]. Still, a number of physicians express concern about tuberculosis in patients awaiting scheduled surgeries, especially in endemic areas where the prevalence is high, such as in Asian countries [21]. As evidenced by the Global Tuberculosis Report of the World Health Organization 2016, Thailand is among the 14 countries with the highest disease severity [21,

22]. Although there are around 120,000 new cases per annum (equivalent to 171 cases per 100,000 of the Thai population), only 55.3% of the cases are detectable. It is therefore still controversial whether chest radiographic examinations form a useful part of preoperative evaluations and should be no longer considered as mandatory by preoperative guidelines.

In an investigation of the causes of unnecessary testing, Brown et. al. found several influencing factors, such as practice traditions, the belief that other physicians want the tests done, medicolegal concerns, a desire to avoid surgical delays or cancellations, and a lack of awareness of evidence and guidelines [23]. However, the study depended on the semi-structured format interview and thus no model on factors affecting preoperative chest radiography request was proposed. We hypothesized that knowing the root causes that resulted in preoperative chest radiography request would aid in developing measures or guidelines to curb unnecessary requests. Our study was designed with the primary objective to determine the factors that influenced medical personnel in the residency and fellowship training programs about the importance and necessity of preoperative chest radiography request in patients undergoing gastrointestinal endoscopy. Secondary objective of the study was to assess the baseline general knowledge of medical personnel on preoperative chest x-ray (CXR) and tuberculosis.

## Materials and methods

### Study design and participants

This cross-sectional clinical study was conducted at a tertiary-care, university-based hospital. Two hundred and sixty-four participants were recruited from the residency and fellowship training programs of the Departments of Anesthesiology (80 individuals), Surgery (171 individuals), and Gastrointestinal Medicine (13 individuals), all under the Faculty of Medicine. The inclusion criteria were medical personnel who agreed to participate and were able to comprehend Thai language. The exclusion criteria were medical personnel who refused to participate in the study. Sample size calculation was performed based on primary objective which was to identify factors contributing to preoperative endoscopic CXR request. The calculation formula for Structural Equation Model (SEM) [24]; anticipated effect size 0.5, desired statistical power level 0.8, number of latent variables 4 and probability level 0.05, recommended minimal sample size (n) of 116.

The conceptual framework was constructed based on previous study on the causes of unnecessary preoperative testing (Fig 1) [23]. The model proposed 3 hypotheses which might reflect the motives of medical personnel toward preoperative CXR request for GI endoscopy. The 3 hypotheses included H1: awareness of patients' history and co-existing diseases (PHC), H2: adherence to the hospital guideline and policy (HGP) and H3: prevention of patient-related complications (PPC). The questionnaire was subsequently generated by 5 specialists from different fields of expertise (namely, anesthesiology, surgery, and psychology) with reference to ASA and The National Institute for Care and Health Excellence (NICE) guidelines for preoperative CXR [5, 7], to assess knowledge and attitudes about preoperative chest radiography of medical personnel. The questionnaire was originally created in Thai language (S1 Appendix; translated to English with the assistance of content expert for publication). The aim of the questionnaire was to determine the factors involved in decision-making by the residents and fellows with regard to their preoperative chest radiography requests. The Index of Item-Objective Congruence (IOC) of the questionnaire was 0.91. Internal consistency of the questionnaire was confirmed by anesthesia alumni and residents from other institutions (Cronbach's alpha coefficient 0.896).

The questionnaire was categorized into 3 main sections. Part 1 dealt with general information on the participants, comprising their age, sex, education, the number of gastrointestinal

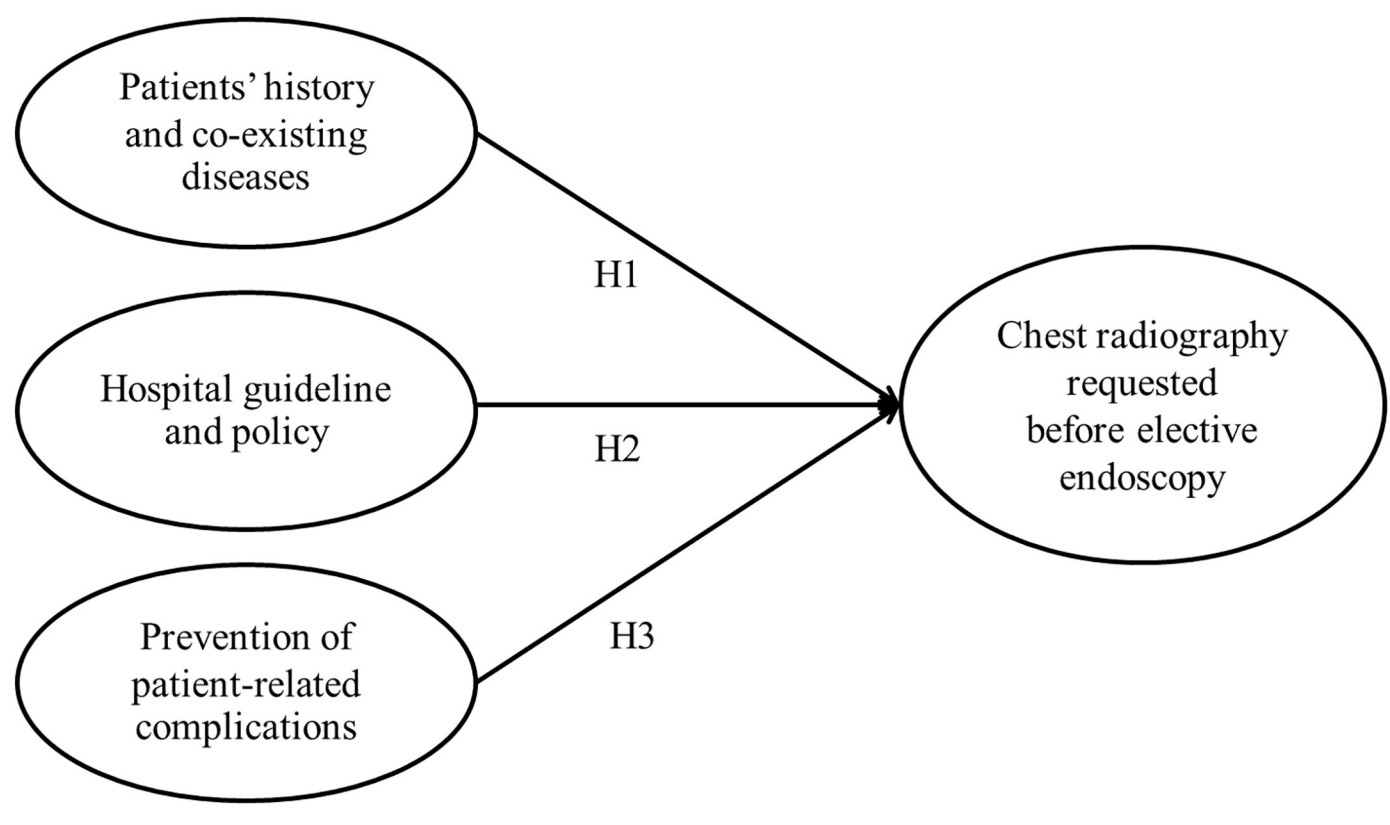

**Fig 1. Model of research.**

endoscopic treatments during the preceding 3 months, responsibility for ordering preoperative CXR, and participation in preoperative CXR evaluation courses. Part 2 focused on the participants' basic knowledge of the necessity and importance of preprocedural or preoperative chest radiography, drawing upon information contained in the guidelines of ASA and NICE, UK. The knowledge test consisted of 18 questions about preoperative CXR for general and ambulatory surgery, CXR indications and knowledge of tuberculosis. Finally, Part 3 addressed the participants' attitudes towards preprocedural or preoperative requests for chest radiography. Each item under the basic knowledge category was a yes–no question, while the items under the attitude category were rated in Likert-scale fashion, such as "extremely important" (5), "very important" (4), "moderately important" (3), "slightly important" (2), and "very slightly important" (1).

The general knowledge questions about preoperative CXR and tuberculosis were categorized into 3 main topics. The first, "patient's age", dealt with the appropriate age for preoperative CXR (question number 9, 10, 12; Q9, Q 10, Q12). The second class of questions, "low-risk surgery and health concerns" (Q14, Q17, Q20), examined the need for CXRs in cases of low-risk surgical procedures or patient health conditions, such as heart disease and asthma. The final category, "TB awareness" (Q24, Q25, Q26), related to the need for preoperative CXRs in particular situations, such as patients with a TB history or as part of TB surveillance activities. Other knowledge questions which did not involve in ambulatory or low-risk surgery were excluded from the analysis. The attitude section consisted of 8 major questions (Q); 4 major questions (Q27-30) comprised 21 sub-questions (Q27.1–27.6, Q28.1–28.5, Q29.1–29.5 and Q30.1–30.5) and 4 other questions without sub-questions (Q31-34). The attitude questions

determined the opinion of the medical personnel toward the necessity of preoperative CXR request and the opinion toward Siriraj pre-anesthesia clinic (SiPAC) recommendation in term of proper CXR request. The attitude questions were categorized into 3 main topics based on proposed model of research (Fig 1) i.e. PHC (Q27.1–27.5), HGP (Q29.4–29.6) and PPC (Q29.1–29.3). The PHC questions contained the question contents similar to those in the knowledge questions but with Likert-scales rather than yes-no answers. Also, a topic related to necessity of chest radiography before elective GI endoscopy following institutional (Siriraj Pre-anesthetic Clinic; SiPAC) guideline or as individual's opinion (CXR) (Q31-34) was regarded as outcome determinant. Other attitudes questions which were not related to endoscopic surgery were excluded from the analysis.

## Data collection

The questionnaires were delivered to the residents and fellows in closed envelopes to the participant's departments of original affiliation in Siriraj Hospital and let the residents or fellows voluntarily participate. Informed consents were obtained from the participants before their entry into the study. A research assistant requested the participants to return the questionnaires within 2 weeks in provided closed containers. If the participants failed to return the questionnaires within the time limit, they would be asked to return the questionnaires one more time. After the participants had completed the questionnaires, the items were calculated and summarized. The protocol for the study was approved by the Siriraj Institutional Review Board, Faculty of Medicine, Siriraj Hospital, Mahidol University, Bangkok, Thailand (Ethical number SI 651/2017).

## Statistical analysis

Descriptive statistic was used to define the demographic characteristics of the respondents by mean, percentage, and standard deviation. Statistical significances of general knowledge score among participants were determined by chi-square test. A p-value of less than 0.05 was regarded as statistically significant. A Confirmatory Factor Analysis (CFA) and a Structural Equation Model (SEM) using the maximum likelihood (ML) method were performed. Internal consistency of factors for CFA was assessed by Cronbach's alpha coefficient. The statistical analyses were performed using SPSS Statistics for Windows, version 18.0 (SPSS Inc., Chicago, Ill., USA). Analysis of Moment Structures (AMOS) version 18.0 was used for CFA and SEM analyses.

## Results

The questionnaires were distributed to the medical personnel during 4 December 2017–30 January 2018. Although 264 medical personnel were recruited to the study, only 142 (53.79%) returned completed questionnaires. Of those respondents, 54 were from the Department of Anesthesiology, 77 from Surgery, and 11 from Gastrointestinal (GI) Medicine. Surgeons represented the major subpopulation group (54.22%), followed by anesthesiologists (38.03%) and gastroenterologists (7.75%). Residents were classified according to year of training. Anesthesiology residents attended a 3-year training program whereas Surgery residents completed 4–5 years of training. Some of the participants were Surgery fellows, the ones who further their studies on surgical sub-specialties. Gastroenterologists were medical personnel who participated in the sub-specialty fellowship training program. Table 1 details the baseline characteristics of the participants (age, gender, position in training programs, preoperative CXR course attendance, being a key individual to order preoperative CXRs, and GI endoscopic treatment experience during the preceding 3 months).

**Table 1. Demographic data of the questionnaire respondents.**

| Variables | Mean ± SD*/n (%) | | | |
|---|---|---|---|---|
| | Anesthesiologists (n = 54) | Surgeons (n = 77) | Gastroenterologists (n = 11) | Total (n = 142) |
| Age (Years) | 28.24 ± 1.44 | 29.17 ± 2.35 | 31.27 ± 1.74 | 28.98 ± 2.15 |
| Sex | | | | |
| Female | 46 (85.2) | 24 (31.2) | 5 (45.5) | 75 (52.8) |
| Male | 8 (14.8) | 53 (68.8) | 6 (54.5) | 67 (47.2) |
| Position | | | | |
| Resident** | 54 (100) | 69 (89.6) | - | 123 (86.6) |
| 1st | 16 (29.6) | 21 (27.3) | - | 42 (29.6) |
| 2nd | 17 (31.5) | 23 (29.9) | - | 46 (32.4) |
| 3rd | 21 (38.9) | 18 (23.4) | - | 39 (27.5) |
| 4th | - | 11 (14.3) | - | 11 (7.7) |
| 5th | - | 4 (5.2) | - | 4 (2.8) |
| Fellow | - | 8 (10.4) | 11 (100) | 19 (13.4) |
| Having attended preoperative CXR course | 26 (48.1) | 36 (46.8) | 5 (45.5) | 68 (47.9) |
| Being a key individual for ordering preoperative Chest x-ray | 24 (44.4) | 71 (92.2) | 11 (100) | 106 (74.6) |
| Being a part of Gastrointestinal endoscopic treatment in the preceding 3 months | | | | |
| 0–10 cases | 42 (77.7) | 50 (64.9) | 1 (9.0) | 93 (65.5) |
| >10 cases | 12 (22.3) | 27 (35.1) | 10 (91.0) | 49 (34.5) |

* SD, Standard deviation

**Classified according to year of training

Data presented as mean ± SD or number (%).

Overall baseline general knowledge on preoperative CXR and tuberculosis among 3 medical specialties did not vary. However, in the case of Q9, which asked whether all patients of any age who are undergoing surgery should have preoperative CXRs, the gastroenterologists had a significantly higher proportion of correct answers (36.4%) than the other two work groups. Another noticeable difference was observed with Q14, which asked whether a patient with non-active asthma requires preoperative chest radiography. For that question (Q14), surgeons had the highest proportion of correct answers (79.2%) (Table 2). The questions that belonged to each topic which was regarded as factor in the conceptual framework (Fig 1) expressed good internal consistency as determined by Cronbach's alpha and were then subjected to Confirmatory Factor Analysis (CFA) (Table 3). The CFA results demonstrated a fitness of data into the hypothetical model: root mean square error of approximation (RMSEA) = 0.063 ($< 0.08$); comparative fit index (CFI) = 0.944 ($\geq 0.90$); goodness of fit index (GFI)_ = 0.906 ($\geq 0.90$); adjusted goodness of fit index (AGFI) = 0.852 ($\geq 0.80$); and Chi-square/df = 1.559 ($< 3$).

Structural Equation Model (SEM) was established to demonstrate the associations between hypothesized factors with the outcome, the intention to request preoperative CXR for elective endoscopic procedures (Fig 2). The SEM was assessed for fitness of data after model adjustment which resulted in acceptable values for all indices: RMSEA = 0.066 ($< 0.08$); CFI = 0.936 ($\geq 0.90$); GFI_ = 0.901 ($\geq 0.90$); AGFI = 0.848 ($\geq 0.80$); and Chi-square/df = 1.918 ($< 3$) (Table 4). Among all hypothesized factors including PHC, HGP and PPC, none of them was statistically associated with an intention for preoperative endoscopic CXR request as determined by the p-value $> 0.05$ (Table 5).

**Table 2. The respondents' scores for the chest x-ray knowledge questions.**

| Topics of general knowledge questions | Respondents with correct answers: n (%) | | | p-value |
|---|---|---|---|---|
| | Anesthesiologists (n = 54) | Surgeons (n = 77) | Gastroenterologists (n = 11) | |
| **Topics related to patient's age** | | | | |
| Q9 Every patient | 4 (7.4) | 23 (29.9) | 4 (36.4) | **0.004***  |
| Q10 Over 45 years old | 48 (88.9) | 68 (88.3) | 10 (90.9) | 0.967 |
| Q12 All pediatric patients | 50 (92.6) | 69 (89.6) | 9 (81.8) | 0.536 |
| **Topics related to low-risk surgery and health concerns** | | | | |
| Q14 Colonoscopy in non-active asthma | 31 (57.4) | 61 (79.2) | 4 (36.4) | **0.002***  |
| Q17 Cataract surgery in CKD requiring hemodialysis | 41 (75.9) | 44 (57.1) | 6 (54.5) | 0.069 |
| Q20 EGD in no underlying disease | 38 (70.4) | 58 (75.3) | 6 (54.5) | 0.342 |
| **Topics related to TB awareness** | | | | |
| Q24 Prevalence of TB | 14 (25.9) | 28 (36.4) | 1 (9.1) | 0.124 |
| Q25 Incidence of TB | 37 (68.5) | 56 (72.7) | 9 (81.8) | 0.649 |
| Q26 Specificity of chest x-ray for TB | 13 (24.1) | 20 (26.0) | 2 (18.2) | 0.848 |

*Significant at $p < 0.05$ by Chi-square test

Abbreviations: Q, Question; CXR, Chest x-ray; CKD, Chronic kidney disease; EGD, Esophagogastroduodenoscopy; TB, Tuberculosis

## Discussion

Our study investigated the factors influencing preprocedural CXR request for gastrointestinal endoscopy among medical personnel. Overall baseline general knowledge on preoperative CXR was comparable between different medical specialties. The validated questionnaire incorporated sets of questions that led to the factors relating to preoperative CXR request. The

**Table 3. Factors used for the Confirmatory Factor Analysis (CFA).**

| Factors | Question Number | Measurement variable | Cronbach's Alpha |
|---|---|---|---|
| Patients' history and co-existing diseases (PHC) | Q27.1 | History of pulmonary tuberculosis | 0.823 |
| | Q27.2 | History of heart disease | |
| | Q27.3 | History of chronic obstructive pulmonary disease | |
| | Q27.4 | History of upper respiratory tract infection | |
| | Q27.5 | Smoking | |
| Hospital guideline and policy (HGP) | Q29.4 | Prevention of prosecution should there be adverse events during the operation | 0.715 |
| | Q29.5 | Following the hospital's policy | |
| | Q27.6 | Healthy patient older than 45 years with no underlying diseases | |
| Prevention of patient-related complications (PPC) | Q29.1 | Avoidance of operation cancellation by surgeons and anesthesiologists | 0.641 |
| | Q29.2 | Tuberculosis surveillance | |
| | Q29.3 | Prevention of risks or complications during the operation | |
| Chest radiography request before elective GI endoscopy (CXR) | Q31 | Do you consider that the current preoperative evaluation guidelines of Siriraj Preanesthetic Clinic (SiPAC) could reduce and prevent complications during the operations | 0.616 |
| | Q32 | Do you consider that following the current SiPAC preoperative evaluation guidelines could prevent cancellations of operations by surgeons and anesthesiologists? | |
| | Q33 | Do general medical personnel strictly follow the SiPAC preoperative evaluation guidelines | |
| | Q34 | Are you concerned about adverse events or complications during an operation if chest radiography for the patient is not available preoperatively | |

Abbreviation: GI, Gastrointestinal; SiPAC, Siriraj Preanesthetic Clinic

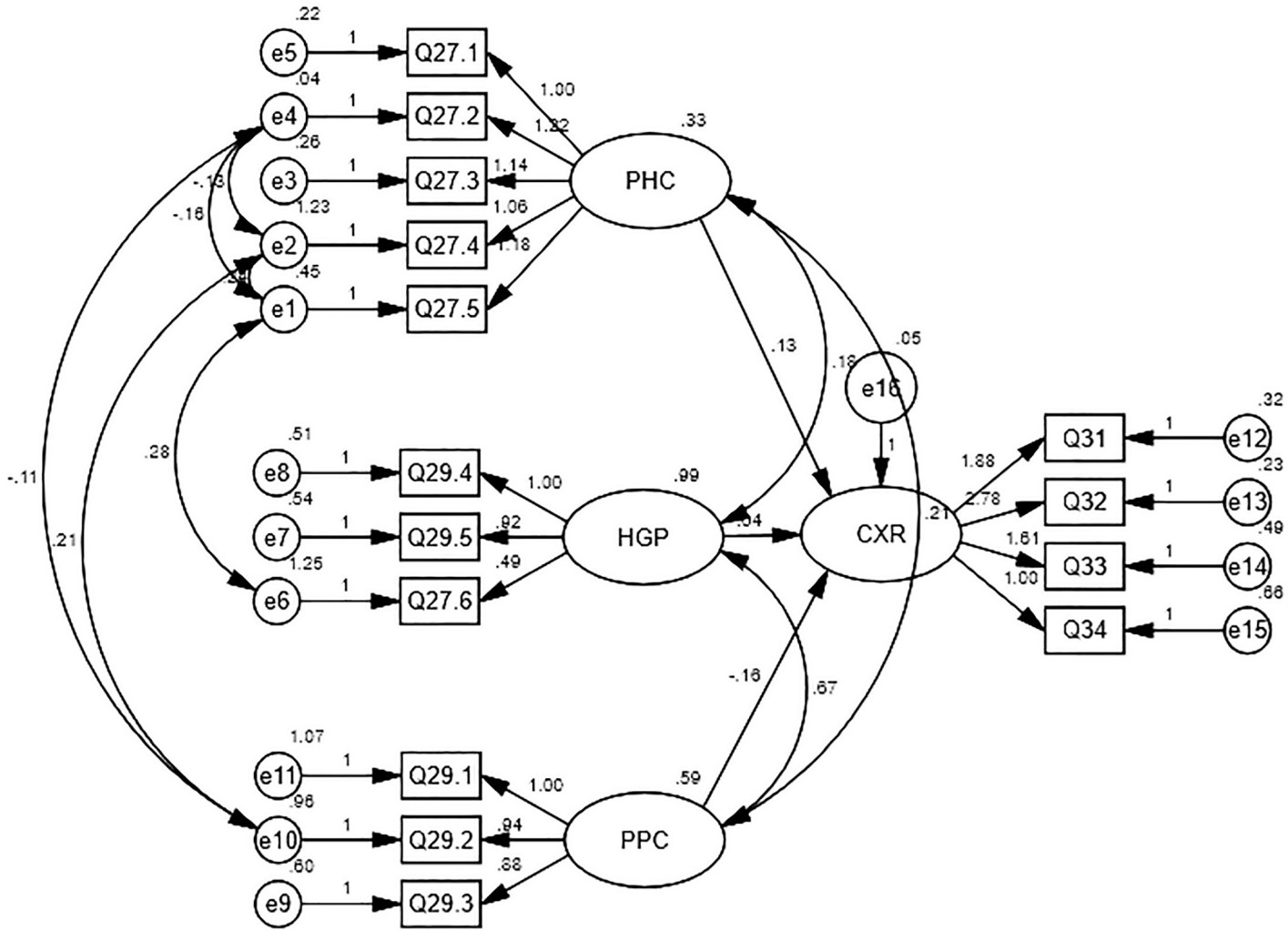

**Fig 2. The Structural Equation Model (SEM) used for analyzing the causal relationships between factors and chest radiography request before elective GI endoscopic procedures.**

factors according to the hypothetical model in association with the intention to perform preoperative CXR for GI endoscopy were validated by CFA and further analyzed with SEM. Nevertheless, no associations between hypothesized factors (PHC, HGP and PPC) and outcome as determined by CXR request (CXR) were observed.

**Table 4. Fit indices for the Structural Equation Model (SEM) before and after adjustment.**

| Index | Criterion | Statistical values obtained from analysis | |
|---|---|---|---|
| | | Before adjustment | After adjustment |
| χ2/df (CMIN/df) | <3 | 2.512 | 1.918 |
| GFI | ≥0.90 | 0.836 | 0.901 |
| AGFI | ≥0.80 | 0.766 | 0.848 |
| CFI | ≥0.90 | 0.832 | 0.936 |
| TLI | ≥0.90 | 0.790 | 0.914 |
| RMSEA | <0.08 | 0.104 | 0.066 |
| **Model summary** | | **Lack of Fit** | **Acceptable Model Fit** |

**Table 5. Hypothesis result of the structural model.**

| Hypothesis | Standardized path coefficients (β) | T-value | p-value | Test result |
|---|---|---|---|---|
| **H1**: Patients' history and co-existing diseases (PHC) => **CXR** | 0.127 | 1.676** | 0.094 | Un-Supported |
| **H2**: Hospital guideline and policy (HGP) => **CXR** | 0.038 | 0.347 | 0.728 | Un-Supported |
| **H3**: Prevention of patient-related complications (PPC) => **CXR** | -0.162 | -0.950 | 0.342 | Un-Supported |

Remark:

***p<0.01,

**p<0.1 and

*p<0.05

It is well documented that preoperative chest radiography, especially in ambulatory settings, might not be essential. This is because only 12% of patients are found to have abnormal CXR findings and, interestingly, as few as 0.03% need further investigation and treatment [9]. Chest radiography is therefore recommended for patients over 60 years of age and is not routinely required for before elective endoscopic procedures [2, 6, 7]. For other minor surgeries, the requirement of preoperative chest radiography could be omitted because it did not alter patient management and rate of postoperative pulmonary complications or adverse events [25–28]. Instead, vigilant history taking and physical examination might be adequate for preoperative preparation [29]. Despite the establishment of international guidelines on preoperative chest radiography over the past 3 decades, there are still no official institutional and Thai national guidelines being issued. In addition, proper guidelines for the preoperative preparation of elective ambulatory cases are yet to be established. As a result, a trend toward unnecessary chest radiography requirement before minor elective operation especially gastrointestinal endoscopy is frequently observed among medical profession. This may be due to the fact that Thailand is an endemic area of tuberculosis as defined by World Health Organization (WHO) [21]. Other possible reasons included intention to complete preoperative preparation or avoidance of case cancellation according to lack of adequate preoperative testing [23]. For these reasons, it would be invaluable to clarify the rationale for preoperative CXR requests. Our study focused on a different population of medical professions—anesthesiologists, surgeons, and gastroenterologists—who had experience with patient evaluations and the requesting of tests preoperatively. We assessed their knowledge and attitudes by analyzing their answers to a validated questionnaire. Taken previous established guidelines and research together [5, 7, 23], we generated a questionnaire that comprised the general knowledge and attitude towards preoperative CXR. Even though our focus was on elective GI endoscopic procedures, other questions irrelevant to endoscopy were present in order to prevent bias when the participants answer the questionnaire.

Referring to the general knowledge questions, most of the surgeons and gastroenterologists were responsible for ordering preoperative CXRs, which might be why they scored higher than anesthesiologists on Q9 (whether all patients undergoing surgery needed preoperative CXRs). Also, surgeons were striking leaders in terms of the percentage of correct answers to Q14 (whether a non-active asthmatic patient requires a preoperative CXR). The medical personnel who had been specially trained in medicine had had a greater chance of encountering tuberculosis patients who, typically, do not show clinical symptoms yet are still contagious. Therefore,

the reason why the gastroenterologists exhibited the lowest percentage of staff with correct answers to the questions related to the prevalence and specificity of CXR for tuberculosis (Q24 and Q26) might be because they are strongly concerned about patients at risk. However, the number of gastroenterologists was much less than the numbers for the staff in the 2 other groups, which is consistent with the limited number of training positions in gastroenterology. Therefore, better results and statistical significance might be achieved with an increased number of gastrointestinal-medicine respondents. The results of our analysis revealed no striking differences in the knowledge of all 3 groups of respondents. Nevertheless importance of TB for preoperative-CXR ordering needed to be highlighted among all medical professions according to relatively lower proportion of participants with correct answers comparing with other topics. The concept of identifying factors responsible for preoperative endoscopic CXR request was to find solutions for our institution to limit the unnecessary test ordering for better cost effectiveness and to reduce risk of radiation exposure. In spite of good model fit for both CFA and SEM, however, the hypothesized factors failed to express association with the outcome meaning that there might be other factors involved.

Our study is the first to find factors for endoscopic CXR request. Similar study regarding an evaluation of factors influencing preoperative testing prior to low-risk surgery was reported previously. The results from retrospective study analysis by Bayesian generalized linear mixed model suggested that institution size was the factor associated with excessive preoperative blood testing. Larger institutions were assumed to have enough resource for routine blood tests and such testing might be general practice of the institutions [30]. This result supported the speculation that there could be other factors besides patients' characteristics and physicians' perspectives that accounted for the behavior.

There were some limitations of the study we would like to address. Apart from the hypothesized factors, other non-clinical influences were not introduced to the framework model. As clinical criteria were regarded as important guidance for clinical decision-making, a variety of non-clinical aspects might be involved such as patient's worries, physician's time constraints and physician's personal experience or belief [31]. Had these non-clinical factors been considered, significant relationship with the wish for preoperative endoscopic CXR might have been achieved. Another limitation was the actual practice of each individual on CXR request was not monitored therefore their actions might not be well predicted. Our future plan is to directly observe the actual practice on preoperative chest x-ray of the medical personnel after their knowledge and attitudes have been evaluated. Future research could determine alternative factors affecting the actual practices relating to preoperative-CXR ordering. Anyhow, knowledge and attitudes should still be properly enhanced by an establishment of official institutional and/or national guidelines.

## Conclusion

The SEM of the conceptual framework was confirmed with good model fit. However, it failed to demonstrate the relationships between our hypothesized factor i.e. PHC, HGP and PPC and intention to request preoperative endoscopic CXR. Other non-clinical factors might be involved thus requiring further study.

## Supporting information

**S1 Appendix. Questionnaire for assessment of knowledge and attitudes of medical personnel about preprocedural or preoperative requests for chest radiography.**
(PDF)

**S1 File. Raw data of knowledge and attitudes about preoperative chest radiography.**
(XLSX)

## Acknowledgments

This study was facilitated by the Integrated Perioperative Geriatric Excellent Research Center, Faculty of Medicine, Siriraj Hospital, Mahidol University, Bangkok, Thailand. The authors are grateful to Assist. Prof. Dr. Chulaluk Komoltri of the faculty's Clinical Epidemiology Unit and Miss Rinrada Preedachitkul of Siriraj Health Policy Unit, Siriraj Hospital, Mahidol University for the statistical analyses.

## Author Contributions

**Conceptualization:** Pawit Somnuke, Cherdsak Iramaneerat, Arunotai Siriussawakul.

**Data curation:** Pawit Somnuke, Rachaneekorn Ramlee, Waratchaya Ratanapaiboon, Passorn Thommaaksorn.

**Formal analysis:** Rachaneekorn Ramlee, Somsit Duangekanong.

**Funding acquisition:** Arunotai Siriussawakul.

**Investigation:** Pawit Somnuke, Rachaneekorn Ramlee, Arunotai Siriussawakul.

**Methodology:** Pawit Somnuke, Rachaneekorn Ramlee, Waratchaya Ratanapaiboon, Passorn Thommaaksorn, Arunotai Siriussawakul.

**Project administration:** Rachaneekorn Ramlee.

**Supervision:** Arunotai Siriussawakul.

**Writing – original draft:** Pawit Somnuke.

**Writing – review & editing:** Arunotai Siriussawakul.

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
