## [Decision Letter · Decision Letter 0]

12 Aug 2020

PONE-D-20-04685

Medical Personnel Knowledge and Attitudes About Preoperative Chest Radiography for Elective Endoscopic Procedures

PLOS ONE

Dear Dr. Siriussawakul,

Thank you for submitting your manuscript to PLOS ONE. After careful consideration, we feel that it has merit but does not fully meet PLOS ONE’s publication criteria as it currently stands. Therefore, we invite you to submit a revised version of the manuscript that addresses the points raised during the review process.

We look forward to receiving your revised manuscript.

Kind regards,

Nancy Beam, PhD

Staff Editor

PLOS ONE

Journal Requirements:

2. In ethics statement in the manuscript and in the online submission form, please provide additional information about the patient records used in your retrospective study. Specifically, please ensure that you have discussed whether all data were fully anonymized before you accessed them and/or whether the IRB or ethics committee waived the requirement for informed consent. If patients provided informed written consent to have data from their medical records used in research, please include this information.

3. Please provide additional details regarding medical personnel consent. In the ethics statement in the Methods and online submission information, please ensure that you have specified (1) whether consent was informed and (2) what type you obtained (for instance, written or verbal, and if verbal, how it was documented and witnessed). If your study included minors, state whether you obtained consent from parents or guardians. If the need for consent was waived by the ethics committee, please include this information.

Reviewers' comments:

Reviewer's Responses to Questions

**Comments to the Author**

1. Is the manuscript technically sound, and do the data support the conclusions?

Reviewer #1: Partly

Reviewer #2: Yes

Reviewer #3: Yes

2. Has the statistical analysis been performed appropriately and rigorously? 

Reviewer #1: No

Reviewer #2: Yes

Reviewer #3: Yes

3. Have the authors made all data underlying the findings in their manuscript fully available?

Reviewer #1: Yes

Reviewer #2: Yes

Reviewer #3: Yes

4. Is the manuscript presented in an intelligible fashion and written in standard English?

Reviewer #1: Yes

Reviewer #2: Yes

Reviewer #3: Yes

5. Review Comments to the Author

Reviewer #1: General comments

- The methods used in this study are mostly sound. In parts, there has been inappropriate use of statistical tests and over-interpretation of results.

- The manuscript is written well and is largely intelligible throughout.

- It is helpful that the authors have included the raw data and the questionnaire used in the study.

- The discussion has new results that are better placed in the results section, does not reference sufficient literature to contextualise the results and leaves the reader without a discussion of the limitations or a conclusion.

- The results are, at times, difficult to read owing to long sentences and ambiguous references to questions in the questionnaire.

- The manuscript could be improved overall by being more explicit about what the main findings were and why these findings are relevant to the field.

- Almost half of the references used in the manuscript are more than 15 years old. Has the literature evolved since then? Have the methods used in pre-operative/pre-endoscopy investigation changed since then? Does this affect the assertions you make in your manuscript? Having conducted a quick literature search, I believe there have been several articles and international guidelines published on this topic over the past 15 years.

Introduction

- Well written. Well done.

- Establishes background, importance of work, problem and niche for solution.

- The wording of “Barely 1% of the tests from overall patients have been reported to reveal abnormalities that would affect perioperative outcomes” is a bit strange to me. My suggestion to improve this would be “Less than one per cent of tests from all patients have been reported to reveal abnormalities that could affect perioperative outcomes”

Methods

- Was the questionnaire validated and/or compared to previously designed questionnaires? If it was validated, then how was it validated?

- What method was used to generate the index of consistency? This should be described in full. Does 0.91 allow for group comparisons and/or individual comparisons? Assuming the use of Cronbach’s alpha, 0.91 would allow individual comparisons over time – have you anonymised but maintained an identifiable code for the participants to do so? It would be worth mentioning the answers to these questions.

- The authors should mention whether the questionnaire was carried out in English alone, English with Thai verbal help, spoken in Thai (with answers captured in English) and/or translated from Thai.

- How did the authors capture whether a respondent believed an attitude item was neither important nor unimportant? How did the authors capture whether a respondent was unsure of an attitude-related item – was this differentiated from a blank answer?

- The attitude section of the questionnaire in the manuscript includes 8 questions with 24 sub-questions but the manuscript says otherwise. Could you please update the manuscript to reflect this, unless I have misunderstood.

- I am very concerned by the high number of statistical comparisons that are being conducted in the subgroup analyses – have you performed a multiple comparisons correction method? If so, which one and why? For example, in table 5 the are approximately 85 comparisons. Using a standard p<0.05 alpha level cut-off would suggest that <4 of those comparisons could be false positives. Four of the total of 8 ‘statistically significant’ results presented in the table may be false positives. This is particularly relevant given that these comparisons are part of a sub-group analysis.

- Why was a Spearman’s rank correlation used for correlation of knowledge and attitudes if this is thought of as a non-parametric measure of correlation? I would suggest using Pearson’s correlation coefficient.

- Please provide the method used for sample size calculation of the retrospective chart review (could be provided as an appendix). The original article is in Thai and I can’t read it, apologies.

- Could you please clarify what you mean by “The preoperative chest radiography statuses of the patients were registered”? What do you mean by status?

- The questionnaire attitudes section gives the five Likert scale responses to answer this question “Do you consider that the current preoperative evaluation guidelines of Siriraj Preanesthetic Clinic (SiPAC) could reduce and prevent complications during the operations?” All possible answers to this question do not make sense – please provide any verbal/written feedback that was given to clarify this.

Results

- Table 1 and 2 notes: “*Significant at p < 0.05”. Please indicate what statistical test was used for this. I suggest you make captions stand-alone, such that they can be read without having to reference other sections (eg. methods).

- Table 1: Did you ask or allow respondents to identify as a non-binary gender? It is important to include these categories, even if there were no respondents that identified. Also, in the sub-group analysis tables you stratify according to sex but asked about gender in the questionnaire. Please update to reflect the information you received.

- In this section of the results, you have re-stated what is inherent in the table. There is no need to ‘double report’ this. Please summarise this or remove it from the text.

o “There were significant mean age differences between the 3 groups of medical professions (anesthesiologists, surgeons, and gastroenterologists), with the highest mean age being for gastroenterologists (31.27 ± 1.74 years), followed by surgeons (29.17 ± 2.35) and anesthesiologists (28.24 ± 1.44). The overall proportions of male and female participants were 47.2% and 52.8%, respectively.”

- The whole paragraph on demographic characteristics is effectively conveyed in the table. My suggestion would be to significantly cut the paragraph on demographic characteristics, for readability.

- The paragraph starting with “The knowledge and attitudes about preoperative chest radiography…” is largely method-related and I believe it would be better suited in the methods section. How does this paragraph differ to the questionnaire information you have detailed in the methods section?

- Were the sub-group analyses identified a priori or were they post hoc? Either way, I believe some justification is necessary as to why the sub-group analyses have been performed and some sort of adjustment for multiple comparisons must be made. As an aside, the other pertinent question is ‘do these analyses add to the paper and its overall message?’ My impression from having read the paper is no – I would suggest you make it more explicit as to why these analyses are important.

- Could you please detail what is meant by “good” and “poor” attitudes? What method was used to assess this?

- The idea of correlating attitude and knowledge scores seems problematic as I am unclear the exact method you have used to do this. Attitudes, while they have been captured here semi-quantitatively, are not quantitative and often knowledge is insufficient to change attitudes. I believe that correlating the two is fundamentally flawed.

- No data is presented in the results section regarding the retrospective patient chart review. Please include this in the results section.

Discussion

- The article would be improved by adding an initial short paragraph to the discussion that summarises the main findings of the study.

- Paragraph 1 of the discussion brings in new results and is better placed in the results section.

- Paragraphs 2 and 3 of the discussion summarise and attempt to rationalise the results. These two paragraphs would be strengthened by i) making the paragraphs more succinct by focusing on the most salient points, ii) by comparing and contrasting your results with the literature (how does your research fit into the broader literature on CXR prior to endoscopy?) and iii) justifying your assertions based on other published literature.

- The authors do not discuss the limitations of their study and the manuscript be strengthened by such a discussion.

- The authors do not provide a concluding paragraph emphasising the importance and utility of this research. The manuscript would be enhanced by this.

o How will this research be used at Mahidol University hospitals? Will there be an intervention carried out to improve the knowledge and attitudes of physicians?

- “This might be because of a better decision-making ability of female respondents” Did you assess decision-making ability? While this may be true, I do not see how your results justify this assertion. Please provide additional reasoning for this.

- Why is “inactive pulmonary infiltration” non-significant? This could represent latent tuberculosis which may significant in certain patient populations. Could you please clarify?

- Limitations

o Understand the knowledge and attitudes of practitioners does not always reliably predict their actions and course of practice. Have they addressed this?

o Low completion rate – why? Will this bias results?

o “Our study focused on a diverse population of medical professions—anesthesiologists, surgeons, and gastroenterologists—who had experience with patient evaluations and the requesting of tests preoperatively.” I disagree that this is a diverse population of medical professions – they are all specialties that have the potential to routinely order, and need knowledge of, endoscopies. Is there any potential bias that could result due to this?

Minor textual amendments:

- Page 5 of manuscript “2,030 outpatient charts produced January–November 2017” should read “2,030 outpatient charts produced between January and November 2017”

- Page 6 of manuscript “others dealt with general, preoperative, CXR indications, and knowledge of tuberculosis.” This sentence is unclear. Could you please clarify it.

- Page 6 of manuscript “The attitude test consisted of 11 questions” should read “The attitude section consisted of 11 questions.”

- Page 11 of manuscript “Our retrospective data indicated that that there” should read “Our retrospective data indicated that there”, delete the extra ‘that’

Reviewer #2: Dear author manuscript is well written but has some minor corrections to be done. the discussion part is however well written but results from other studies were not compared to the present study either similar / dissimilar.

Reviewer #3: Thank you for the opportunity to review this manuscript. The advantage of this study was conducted scientifically and systematically. The manuscript is well written and described clearly. Several suggestions to improve this manuscript are listed below.

The methods section of the abstract needs to contain more information, especially regarding the study location and duration. It will reflect the inference population. Research tool (the questionnaire) needs to be elaborated further as to whether a newly developed questionnaire or a validated questionnaire was used.

In this study, two study designs were employed, a prospective clinical study and an observational retrospective study. Is there any reason why it cannot be called a cross-sectional study since all the information is obtained at a single point of time with no element of follow-up?

What is the basis of taking 264 participants? Was it based on any sample size calculation? In this study, elaborate on how a sample size of 264 was determined. Any exclusion criteria for study participants? What does it mean by "incomplete medical records"? Who decides whether an x-ray is uninterpretable? Include a reference number for ethical clearance.

Define "resident" and "fellow".

This study uses a newly developed questionnaire to assess knowledge and attitude towards preoperative chest radiography for elective endoscopic procedures. It is important that the questionnaire is validated before it is being used for the actual study. Furthermore, the readers must be convinced that the questionnaire undergoes a proper questionnaire validation process as it will determine the validity of the results. "The Index of consistency score of the questionnaire was 0.91". Does this indicate internal consistency reliability? The assessment of content validity needs further elaboration (e.g. assessment of content validity index by a group of independent experts). Any reason why factor analysis (EFA and CFA) was not performed?

Elaborate sampling method. Any method employed to ensure the representativeness of the study participants in terms of the department and work experience?

What does it mean by "Statistical significances were determined by chi-squared tests"? Statistical significance refer to the p-value and can be obtained from all statistical test (not just the chi-squared test).

Rephrase "The data with a normal distribution were analyzed and reported as percentage, mean, and standard deviation". All data were analyzed. Data with normal distribution needs to be presented as mean and its standard deviation, whereas data with skewed distribution needs to be resented as median and interquartile range. Frequency and its percentage are used to describe categorical variables.

The sample size calculation needs to be placed under "study design and participants" and not under the method of statistical analysis.

The knowledge test consisted of 18 questions; 7 were about surgery in ambulatory settings, whereas the others dealt with general, preoperative, CXR indications, and knowledge of tuberculosis. The attitude test consisted of 11 questions. Elaborate method to obtain the total score for each domain.

The response rate in this study is low (53.79%). In terms of statistical power, was it still adequate? Discuss the response rate in relation to external validity of this study.

The description regarding the questionnaire and the methods of scoring for each item (paragraph two of the results section) needs to be placed under the methods section.

Lengthy text presentation of the statistical results. Most of the information can be obtained from the tables, and presenting the results in both text and tables are redundant.

In table 1, 2, indicate the statistical analysis conducted to obtain each p-value. Here, the comparison of numerical variables between three independent groups was conducted using one-way ANOVA, and for categorical variables using either the Chi-squared test or Fisher exact test. Methods of these statistical analyses need to be included in the methods section.

6. PLOS authors have the option to publish the peer review history of their article (what does this mean?). If published, this will include your full peer review and any attached files.

Reviewer #1: **Yes: **Andrés Noé

Reviewer #2: **Yes: **Aisha Wali

Reviewer #3: No

---

## [Author Response · Author response to Decision Letter 0]

24 Sep 2020

Manuscript number: PONE-D-20-04685

Journal: PLOS ONE

Title: Medical Personnel Knowledge and Attitudes About Preoperative Chest Radiography for Elective Endoscopic Procedures

Dear editor,

Thank you for giving a chance to improve our manuscript. Hopefully, you would appreciate our revised version. We described the change in response to the reviewers and in the manuscript.

Regards

Arunotai Siriussawakul

Associate professor 

Department of Anesthesiology,

Siriraj Hospital, Mahidol University, Thailand.

 

Journal Requirements:

Answer: We did as suggestion.

2. In ethics statement in the manuscript and in the online submission form, please provide additional information about the patient records used in your retrospective study. Specifically, please ensure that you have discussed whether all data were fully anonymized before you accessed them and/or whether the IRB or ethics committee waived the requirement for informed consent. If patients provided informed written consent to have data from their medical records used in research, please include this information.

Answer: We have decided not to include this set of data in the manuscript because it did not add to our primary or secondary objective of the study. Also, part of the data was discussed in our previous publication which was mentioned in the introduction part as ref 20.

3. Please provide additional details regarding medical personnel consent. In the ethics statement in the Methods and online submission information, please ensure that you have specified (1) whether consent was informed and (2) what type you obtained (for instance, written or verbal, and if verbal, how it was documented and witnessed). If your study included minors, state whether you obtained consent from parents or guardians. If the need for consent was waived by the ethics committee, please include this information.

Answer: The research assistance applied the questionnaires in closed enveloped to the residents and fellows at their departments of affiliation. The document explaining the details and objectives of the study was attached with the questionnaire. The persons who would like to participate voluntarily took the questionnaire. They were asked to return the questionnaires within 2 weeks in provided closed containers. If not, they would be asked to return the questionnaires again by the research assistance. The IRB committee had agreed to let us proceed with the study (ethic no. Si 651/2017). 

Reviewers' comments:

Reviewer's Responses to Questions

Comments to the Author

1. Is the manuscript technically sound, and do the data support the conclusions?

Reviewer #1: Partly

Reviewer #2: Yes

Reviewer #3: Yes

2. Has the statistical analysis been performed appropriately and rigorously?

Reviewer #1: No

Reviewer #2: Yes

Reviewer #3: Yes

3. Have the authors made all data underlying the findings in their manuscript fully available?

Reviewer #1: Yes

Reviewer #2: Yes

Reviewer #3: Yes

4. Is the manuscript presented in an intelligible fashion and written in Standard English?

Reviewer #1: Yes

Reviewer #2: Yes

Reviewer #3: Yes

5. Review Comments to the Author

Reviewer #1: General comments

- The methods used in this study are mostly sound. In parts, there has been inappropriate use of statistical tests and over-interpretation of results.

Answer: We have revised the data and used appropriate analysis as suggestion.

- The manuscript is written well and is largely intelligible throughout.

Answer: Thank you very much. 

- It is helpful that the authors have included the raw data and the questionnaire used in the study.

Answer: Thank you.

- The discussion has new results that are better placed in the results section, does not reference sufficient literature to contextualize the results and leaves the reader without a discussion of the limitations or a conclusion.

Answer: We have removed that new result from the result section and added details including limitations and conclusion to the discussion section. Also, more related literatures were added.

- The results are, at times, difficult to read owing to long sentences and ambiguous references to questions in the questionnaire.

Answer: We have combined the tables with legends and supplementary appendix to the manuscript. Hopefully it will ease reviewing the result section.

- The manuscript could be improved overall by being more explicit about what the main findings were and why these findings are relevant to the field.

Answer: We have made new analysis which pointed out the factors that reflected intention for preoperative endoscopic chest radiography request.

- Almost half of the references used in the manuscript are more than 15 years old. Has the literature evolved since then? Have the methods used in pre-operative/pre-endoscopy investigation changed since then? Does this affect the assertions you make in your manuscript? Having conducted a quick literature search, I believe there have been several articles and international guidelines published on this topic over the past 15 years.

Answer: We have added updated references as suggestion.

Introduction

- Well written. Well done.

Answer: Thank you very much. We appreciate your comment.

- Establishes background, importance of work, problem and niche for solution.

Answer: Thank you.

- The wording of “Barely 1% of the tests from overall patients have been reported to reveal abnormalities that would affect perioperative outcomes” is a bit strange to me. My suggestion to improve this would be “Less than one per cent of tests from all patients have been reported to reveal abnormalities that could affect perioperative outcomes”

Answer: We did as suggestion. 

Methods

- Was the questionnaire validated and/or compared to previously designed questionnaires? If it was validated, then how was it validated?

Answer: Thank you for your nice suggestion. The questionnaire was newly created. We validated the questionnaire before the data collection by Index of item-objective congruence (IOC) by 5 content experts in the field. We also test the reliability using Cronbach’s alpha assessment from Anesthesia alumni and residents from other institutions which were not included in the study. The details are mentioned in paragraph 2 of the methods section 

- What method was used to generate the index of consistency? This should be described in full. Does 0.91 allow for group comparisons and/or individual comparisons? Assuming the use of Cronbach’s alpha, 0.91 would allow individual comparisons over time – have you anonymised but maintained an identifiable code for the participants to do so? It would be worth mentioning the answers to these questions.

Answer: Actually, we validated the questionnaire in this study by Index of Item-Objective Congruence (IOC) by 5 experts in the field. The IOC was 0.91 therefore good content validity was achieved. We had as well assessed the internal consistence of the questionnaire by using Cronbach’s alpha which was 0.896 (good reliability). We have already mentioned this in the materials and methods section. 

- The authors should mention whether the questionnaire was carried out in English alone, English with Thai verbal help, spoken in Thai (with answers captured in English) and/or translated from Thai.

Answer: The original questionnaire was written in Thai. We translated it to English for publication with the assistance of the experts in the field.

- How did the authors capture whether a respondent believed an attitude item was neither important nor unimportant? How did the authors capture whether a respondent was unsure of an attitude-related item – was this differentiated from a blank answer?

Answer: We agreed with the points you raised. Attitude questions were in Likert pattern thus it would be quite difficult to determine which ones were important or unimportant. Therefore, we re-analyzed the data by using confirmatory factor analysis (CFA) and constructed structural equation model (SEM) to find out the factors that affected the preoperative chest x-ray request of medical personnel.

- The attitude section of the questionnaire in the manuscript includes 8 questions with 24 sub-questions but the manuscript says otherwise. Could you please update the manuscript to reflect this, unless I have misunderstood.

Answer: We have made a correction. Original attitude questionnaire comprised 8 major questions. We, however, picked only 11 sub-questions and 4 major questions that related to elective gastrointestinal endoscopic procedures and tuberculosis awareness. We have clearly elaborated this in the methods. 

- I am very concerned by the high number of statistical comparisons that are being conducted in the subgroup analyses – have you performed a multiple comparisons correction method? If so, which one and why? For example, in table 5 there are approximately 85 comparisons. Using a standard p<0.05 alpha level cut-off would suggest that <4 of those comparisons could be false positives. Four of the total of 8 ‘statistically significant’ results presented in the table may be false positives. This is particularly relevant given that these comparisons are part of a sub-group analysis.

Answer: We agreed that sub-group analyses might well create false positive results therefore we decided to remove them from the result section. Additionally, we re-analyzed the data with more appropriate statistics method according to the objectives of the study.

- Please provide the method used for sample size calculation of the retrospective chart review 

Answer: We decided not to include this information because it did not answer the research question. In addition, our members in the research group have used part of this information in the publication which is cited in this study.

- Could you please clarify what you mean by “The preoperative chest radiography statuses of the patients were registered”? What do you mean by status?

Answer: We decided not to include this set of data in the manuscript because it did not match with the objectives of the study.

- The questionnaire attitudes section gives the five Likert scale responses to answer this question “Do you consider that the current preoperative evaluation guidelines of Siriraj Preanesthetic Clinic (SiPAC) could reduce and prevent complications during the operations?” All possible answers to this question do not make sense – please provide any verbal/written feedback that was given to clarify this.

Answer: The Siriraj Preanesthetic Care Unit (SiPAC) was set up in our institution to provide preoperative evaluation and preparation according to the medical condition of the patient as well as laboratory investigation. This setting is related to the hospital policy to reduce perioperative risk and improve cost effectiveness of patient treatment. However, the system for evaluation of hospital economy in term of patient testing for preoperative preparation is yet to be established. The question arises whether there is a policy for reduction of unnecessary laboratory testing including chest radiography. We, therefore, constructed the research framework the identify factors that influence chest x-ray request.

Results

- Table 1 and 2 notes: “*Significant at p < 0.05”. Please indicate what statistical test was used for this. I suggest you make captions stand-alone, such that they can be read without having to reference other sections (eg. methods).

Answer: The tables were re-designed and rearranged in the new format. The footnote stated the statistics used and the abbreviations in the table. At present, table is 1 the demographic characteristics of the patients. The results are reported as descriptive statistics with mean±SD or number (%). Table 2 displays the baseline knowledge about preoperative chest x-ray among medical personnel. The significance is determined by p < 0.05 using Chi-square. We do hope these will make the results more understandable.

- Table 1: Did you ask or allow respondents to identify as a non-binary gender? It is important to include these categories, even if there were no respondents that identified. Also, in the sub-group analysis tables you stratify according to sex but asked about gender in the questionnaire. Please update to reflect the information you received.

Answer: All participant (142 persons) identified themselves as female (75 persons) or male (67 persons). We have already changed the word “gender” to sex in the questionnaire.

- In this section of the results, you have re-stated what is inherent in the table. There is no need to ‘double report’ this. Please summarise this or remove it from the text. 

o “There were significant mean age differences between the 3 groups of medical professions (anesthesiologists, surgeons, and gastroenterologists), with the highest mean age being for gastroenterologists (31.27 ± 1.74 years), followed by surgeons (29.17 ± 2.35) and anesthesiologists (28.24 ± 1.44). The overall proportions of male and female participants were 47.2% and 52.8%, respectively.”

Answer: We have removed this from the text as suggestion.

- The whole paragraph on demographic characteristics is effectively conveyed in the table. My suggestion would be to significantly cut the paragraph on demographic characteristics, for readability.

Answer: According to the fact that table 1 had already demonstrated complete demographic data of the participants, we decided to remove unnecessary paragraph on demographic characteristics as suggestion.

- The paragraph starting with “The knowledge and attitudes about preoperative chest radiography…” is largely method-related and I believe it would be better suited in the methods section. How does this paragraph differ to the questionnaire information you have detailed in the methods section?

Answer: It is not different from what is written in methods section. We have moved the paragraph to the methods section as suggestion.

- Were the sub-group analyses identified a priori or were they post hoc? Either way, I believe some justification is necessary as to why the sub-group analyses have been performed and some sort of adjustment for multiple comparisons must be made. As an aside, the other pertinent question is ‘do these analyses add to the paper and its overall message?’ My impression from having read the paper is no – I would suggest you make it more explicit as to why these analyses are important.

Answer: The sub-group was post-hoc analysis. However, we agreed that they might not add to the paper. We decided to remove the sub-group analysis and perform confirmatory factor analysis (CFA) and constructed structural equation model (SEM) as mentioned later on. The demographic data of the participants were reported descriptively.

- Could you please detail what is meant by “good” and “poor” attitudes? What method was used to assess this?

Answer: Attitude questions were in Likert pattern thus it would be quite difficult to determine which ones were poor or good. Therefore, we re-analyzed the data by using confirmatory factor analysis (CFA) and constructed structural equation model (SEM) to find out the factors that affected the chest x-ray request of medical personnel. By SEM analysis we thought it would be more appropriate for the analysis of factors that affected chest x-ray request of medical personnel. The information from SEM would be better to obtain the important data on what was the key element for appropriate CXR request. 

- The idea of correlating attitude and knowledge scores seems problematic as I am unclear the exact method you have used to do this. Attitudes, while they have been captured here semi-quantitatively, are not quantitative and often knowledge is insufficient to change attitudes. I believe that correlating the two is fundamentally flawed. 

Answer: We agreed with your suggestion that the idea of correlating knowledge and attitudes may not be appropriate. Therefore, we decided separate the questionnaire into 2 sets including 1. Knowledge part which was later analyzed and reported as demographic data and 2. Attitude part containing questions with scaled items to perform confirmatory factor analysis (CFA) of the measured variables derived from the questionnaire. Subsequently, we generated structural equation model (SEM) to find relationships between the latent variables with the outcome which was intention of CXR request for preprocedural gastrointestinal endoscopic procedures. 

- No data is presented in the results section regarding the retrospective patient chart review. Please include this in the results section.

Answer: We have removed this from the manuscript

Discussion

- The article would be improved by adding an initial short paragraph to the discussion that summarises the main findings of the study.

Answer: We have added summary of main findings as suggestion.

- Paragraph 1 of the discussion brings in new results and is better placed in the results section.

Answer: We have removed the paragraph 1 from the study because it is actually a part of our published data which we have lately added to the introduction (ref. 19). 

- Paragraphs 2 and 3 of the discussion summarise and attempt to rationalise the results. These two paragraphs would be strengthened by i) making the paragraphs more succinct by focusing on the most salient points, ii) by comparing and contrasting your results with the literature (how does your research fit into the broader literature on CXR prior to endoscopy?) and iii) justifying your assertions based on other published literature.

Answer: We did as suggestion.

- The authors do not discuss the limitations of their study and the manuscript be strengthened by such a discussion.

Answer: We added the limitation accordingly.

- The authors do not provide a concluding paragraph emphasising the importance and utility of this research. The manuscript would be enhanced by this. 

How will this research be used at Mahidol University hospitals? 

Answer: We have added the paragraph “The concept of identifying factors responsible for preoperative endoscopic CXR request was to find solutions for our institution to limit the unnecessary test ordering for better cost effectiveness and reduced risk of radiation exposure.” in the discussion section. 

Will there be an intervention carried out to improve the knowledge and attitudes of physicians?

Answer: Future plans will be providing proper knowledge and attitudes to medical personnel and closely follow up on each individual on real practice on preoperative chest x-ray ordering for the patients. We expect that with good knowledge and attitude, unnecessary chest x-ray ordering will tend to decline. However, the true factors influencing chest x-ray request need to be identified.

- “This might be because of a better decision-making ability of female respondents” Did you assess decision-making ability? While this may be true, I do not see how your results justify this assertion. Please provide additional reasoning for this.

Answer: We did not assess decision-making ability in this study. It was just our speculation. We agreed with your suggestion that the results did not indicate the decision-making ability therefore we removed the sentence form the discussion. This could be the one of the limitations of the study in the sense that we could not identify real practice of each individual therefore his/her decision making would not be revealed.

- Why is “inactive pulmonary infiltration” non-significant? This could represent latent tuberculosis which may significant in certain patient populations. Could you please clarify?

Answer: We have deleted this from the manuscript.

Limitations

o Understand the knowledge and attitudes of practitioners does not always reliably predict their actions and course of practice. Have they addressed this?

Answer: We were well aware of this point. Our future plan is directly observing the actual practice on preoperative chest x-ray of the medical personnel. We expect to see minimal unnecessary chest x-ray ordering in the ones with good knowledge and attitudes.

o Low completion rate – why? Will this bias results?

Answer: The participants volunteered to answers the questionnaires. We could not force the ones who refused to participate according to the ethics we declared to the IRB committee. Having said that, based on sample calculation in Structural Equation Mode (SEM) analysis, the minimal sample size required for reliable outcome is 116 (ref.23). We were able to get 142 respondents which was enough for analysis.

o “Our study focused on a diverse population of medical professions—anesthesiologists, surgeons, and gastroenterologists—who had experience with patient evaluations and the requesting of tests preoperatively.” I disagree that this is a diverse population of medical professions – they are all specialties that have the potential to routinely order, and need knowledge of, endoscopies. Is there any potential bias that could result due to this?

Answer: We agreed and so we change the word “diverse” to different.

Minor textual amendments:

- Page 5 of manuscript “2,030 outpatient charts produced January–November 2017” should read “2,030 outpatient charts produced between January and November 2017”

Answer: The sentence was deleted because we did not include this result in the manuscript.

- Page 6 of manuscript “others dealt with general, preoperative, CXR indications, and knowledge of tuberculosis.” This sentence is unclear. Could you please clarify it.

Answer: The sentenced was deleted.

- Page 6 of manuscript “The attitude test consisted of 11 questions” should read “The attitude section consisted of 11 questions.”

Answer: We have already changed the word “test” to section.

- Page 11 of manuscript “Our retrospective data indicated that that there” should read “Our retrospective data indicated that there”, delete the extra ‘that’

Answer: The sentenced was re-written.

Reviewer #2:

Dear author manuscript is well written but has some minor corrections to be done. the discussion part is however well written but results from other studies were not compared to the present study either similar / dissimilar.

Answer: Thank you very much. We have re-written the discussion paragraph comparing our study with others.

Reviewer #3: 

Thank you for the opportunity to review this manuscript. The advantage of this study was conducted scientifically and systematically. The manuscript is well written and described clearly. Several suggestions to improve this manuscript are listed below.

The methods section of the abstract needs to contain more information, especially regarding the study location and duration. It will reflect the inference population. Research tool (the questionnaire) needs to be elaborated further as to whether a newly developed questionnaire or a validated questionnaire was used.

Answer: We added more details in the Materials and Methods section under Study design and participants and Data collection parts.

In this study, two study designs were employed, a prospective clinical study and an observational retrospective study. Is there any reason why it cannot be called a cross-sectional study since all the information is obtained at a single point of time with no element of follow-up?

Answer: The main study is a cross-sectional clinical study. We have removed the retrospective data out of the study because it was a part of our recently published data (we added this as ref.19 and 20 in the introduction section ) which did not add to our result interpretation. 

What is the basis of taking 264 participants? Was it based on any sample size calculation? In this study, elaborate on how a sample size of 264 was determined. Any exclusion criteria for study participants? What does it mean by "incomplete medical records"? 

Answer: The number 264 was from total medical personnel from residency and fellowship training programs in Siriraj Hospital, Mahidol University. Based on our analysis of the Structural Equation Model (SEM), the minimal sample size which resulted in reliable outcome was 116 (ref. 24) as explained in the Materials and methods section, study design and participants part. Exclusion criteria were medical personnel who refuse to participate and the ones who did not comprehend Thai given that original questionnaire was written in Thai. However, the questionnaires were sent to all potential participants because we thought that medical personnel might be busy and only some of them could dedicate time to participate. The phrase “incomplete medical records” was not mentioned when we re-wrote the manuscript.

Who decides whether an x-ray is uninterpretable? 

Answer: We have deleted this from the manuscript.

Include a reference number for ethical clearance.

Answer: We have added the refence number as suggestion.

Define "resident" and "fellow".

Answer: We have made an explanation in paragraph 1 of the result section.

This study uses a newly developed questionnaire to assess knowledge and attitude towards preoperative chest radiography for elective endoscopic procedures. It is important that the questionnaire is validated before it is being used for the actual study. Furthermore, the readers must be convinced that the questionnaire undergoes a proper questionnaire validation process as it will determine the validity of the results. "The Index of consistency score of the questionnaire was 0.91". Does this indicate internal consistency reliability? The assessment of content validity needs further elaboration (e.g. assessment of content validity index by a group of independent experts). Any reason why factor analysis (EFA and CFA) was not performed?

Answer: We have re-analyzed the data. The details are explained in the Materials and methods section, paragraph 2. CFA and SEM were also performed as mentioned in paragraph 2 and 3 of result section. We also invited co-author, Dr. Somsit Daungekanong, who is an expert on statistical analysis to assist with the re-analysis and data interpretation.

Elaborate sampling method. Any method employed to ensure the representativeness of the study participants in terms of the department and work experience?

Answer: We agreed to what you suggested. Anyway, we only have 264 trainees in our institution. Therefore, we decided to use the overall results to guide for the improvement of practice on CXR request and official institution guideline establishment in the future.

What does it mean by "Statistical significances were determined by chi-squared tests"? Statistical significance refer to the p-value and can be obtained from all statistical test (not just the chi-squared test).

Answer: We have re-analyzed the data and used Chi-square test only in Table 2. 

Rephrase "The data with a normal distribution were analyzed and reported as percentage, mean, and standard deviation". All data were analyzed. Data with normal distribution needs to be presented as mean and its standard deviation, whereas data with skewed distribution needs to be resented as median and interquartile range. Frequency and its percentage are used to describe categorical variables.

Answer: We re-analyzed the data and employed descriptive statistics for the demographic characteristics of the medical personnel as shown in Table 1. The results were described by mean±SD and percentage. 

The sample size calculation needs to be placed under "study design and participants" and not under the method of statistical analysis.

Answer: We have moved the sample calculation to the study design and participants as suggestion.

The knowledge test consisted of 18 questions; 7 were about surgery in ambulatory settings, whereas the others dealt with general, preoperative, CXR indications, and knowledge of tuberculosis. The attitude test consisted of 11 questions. Elaborate method to obtain the total score for each domain.

Answer: We have re-analyzed the data for better result interpretation and rationale of the study. We adopted IOC and Cronbach’s alpha to determine the validity and reliability of the questionnaire. With the use of Daniel Soper’s equation model analysis, we were able to obtain the good fit of the model to define the relationship between latent variables and the chest x-ray request (outcome). 

The response rate in this study is low (53.79%). In terms of statistical power, was it still adequate? Discuss the response rate in relation to external validity of this study.

Answer: Actually, based on sample size calculation for SEM (ref. 24), the number of participants we obtained was enough for analysis. Even though the response rate was 53.79%, we did not adopt the result from the analysis to determine the knowledge and attitudes of other groups of medical personnel. Instead, we aimed to use the analysis for an improvement of preoperative evaluation study for proper chest x-ray request in the future.

The description regarding the questionnaire and the methods of scoring for each item (paragraph two of the results section) needs to be placed under the methods section.

Answer: According to the re-analysis of the data for better result interpretation with more appropriate statistics use, we had removed this part.

Lengthy text presentation of the statistical results. Most of the information can be obtained from the tables, and presenting the results in both text and tables are redundant.

Answer: We have cut the redundant phrases out from the results section.

In table 1, 2, indicate the statistical analysis conducted to obtain each p-value. Here, the comparison of numerical variables between three independent groups was conducted using one-way ANOVA, and for categorical variables using either the Chi-squared test or Fisher exact test. Methods of these statistical analyses need to be included in the methods section.

Answer: We did as suggestion.

6. PLOS authors have the option to publish the peer review history of their article (what does this mean?). If published, this will include your full peer review and any attached files.

Do you want your identity to be public for this peer review? For information about this choice, including consent withdrawal, please see our Privacy Policy.

Reviewer #1: Yes: Andrés Noé

Reviewer #2: Yes: Aisha Wali

Reviewer #3: No

**We would like the express our sincere gratitude toward your invaluable reviews. We have made some changes as follows;

1. The article name is changed to “Factors influencing preoperative chest radiography request for elective endoscopic procedures among medical personnel” considering that it reflected our primary objective of the study.

2. We have added to the methods that Hypothetical framework of possible factors that influenced medical personnel of chest x-ray request was constructed. The model validity was confirmed by Confirmatory Factor Analysis (CFA) and subsequently subjected to Structural Equation Model (SEM) to determine any associations. By doing so, we are affirmed that the statistical analysis of variables was appropriate as you have suggested.

3. General knowledge on preoperative chest x-ray was compared among medical personnel by using Chi-square test.

4. The multiple comparisons of subgroup analyses of knowledge and attitudes among medical personnel were removed according to possible false positive results as the reviewer suggested.

5. Correlation between knowledge and attitude by Spearmen’s rank was removed. Instead we determined the association of hypothesized factor with the intention to request for chest x-rays by SEM as mentioned earlier.

---

## [Decision Letter · Decision Letter 1]

28 Oct 2020

Factors influencing preoperative chest radiography request for elective endoscopic procedures among medical personnel

PONE-D-20-04685R1

Dear Dr. Siriussawakul,

We’re pleased to inform you that your manuscript has been judged scientifically suitable for publication and will be formally accepted for publication once it meets all outstanding technical requirements.

Kind regards,

Sanjiv Mahadeva, MRCP, MD

Academic Editor

PLOS ONE

Additional Editor Comments (optional):

The authors have made substantive revision based on the original 3 reviewers comments - these revisions are satisfactory

Reviewers' comments:

Reviewer's Responses to Questions

**Comments to the Author**

1. If the authors have adequately addressed your comments raised in a previous round of review and you feel that this manuscript is now acceptable for publication, you may indicate that here to bypass the “Comments to the Author” section, enter your conflict of interest statement in the “Confidential to Editor” section, and submit your "Accept" recommendation.

Reviewer #2: All comments have been addressed

2. Is the manuscript technically sound, and do the data support the conclusions?

Reviewer #2: Yes

3. Has the statistical analysis been performed appropriately and rigorously? 

Reviewer #2: Yes

4. Have the authors made all data underlying the findings in their manuscript fully available?

Reviewer #2: Yes

5. Is the manuscript presented in an intelligible fashion and written in standard English?

Reviewer #2: Yes

6. Review Comments to the Author

Reviewer #2: (No Response)

7. PLOS authors have the option to publish the peer review history of their article (what does this mean?). If published, this will include your full peer review and any attached files.

Reviewer #2: **Yes: **Aisha Wali

---

## [Editor Report · Acceptance letter]

4 Nov 2020

PONE-D-20-04685R1 

Factors influencing preoperative chest radiography request for elective endoscopic procedures among medical personnel 

Dear Dr. Siriussawakul:

I'm pleased to inform you that your manuscript has been deemed suitable for publication in PLOS ONE. Congratulations! Your manuscript is now with our production department. 

Kind regards, 

on behalf of

Dr. Sanjiv Mahadeva 

Academic Editor

PLOS ONE